# Watching Alkaline Phosphatase Catalysis Through Its Vibrational Fingerprint

**DOI:** 10.3390/biology15010068

**Published:** 2025-12-30

**Authors:** Margherita Tamagnini, Haoyue Jiang, Liana Klivansky, Carlos Bustamante, Alessandra Lanzara

**Affiliations:** 1Department of Molecular and Cell Biology, University of California, Berkeley, CA 94720, USA; 2Applied Science Technology, University of California, Berkeley, CA 94720, USA; 3Materials Sciences Division, Lawrence Berkeley National Laboratory, Berkeley, CA 94720, USA; 4Molecular Foundry, Lawrence Berkeley National Laboratory, Berkeley, CA 94720, USA; 5Institute for Quantitative Biosciences (QB3), University of California, Berkeley, CA 94720, USA; 6Department of Physics, University of California, Berkeley, CA 94720, USA; 7Kavli Nanoscience Energy Institute, University of California, Berkeley, CA 94720, USA

**Keywords:** Fourier transform infrared spectroscopy (FTIR), alkaline phosphatase (ALP), spectral analysis, p-nitrophenyl phosphate (PNPP), molecular fingerprinting

## Abstract

Alkaline phosphatase is an important enzyme involved in many biological processes and is widely studied as a model system for enzyme catalysis. While its structure and activity have been extensively characterized, much less is known about how molecular vibrations evolve during the catalytic reaction. In this work, we use infrared spectroscopy to monitor in real time the conversion of p-nitrophenyl phosphate to p-nitrophenol by alkaline phosphatase in aqueous solution. By comparing time-resolved spectra with reference spectra of the enzyme, substrate, and products, we directly follow product formation and changes in vibrational features during catalysis. This approach provides a direct spectroscopic view of enzymatic activity and illustrates how infrared spectroscopy can be used to study catalytic processes under near-native conditions.

## 1. Introduction

Alkaline phosphatase (ALP) is a key metalloenzyme involved in critical biochemical pathways, playing a key role in metabolism, bone mineralization and diagnostics [1,2,3,4,5,6,7,8]. Clinically, ALP serves as an important biomarker in both bone and chronic kidney disease (CKD) [6,7,8,9]. In bone disorders, circulating ALP—particularly the bone-specific isoform—reflects osteoblastic activity and bone turnover and is routinely used to diagnose and monitor metabolic bone diseases [10,11,12]. In CKD, elevated ALP levels are associated with impaired mineral metabolism and increased cardiovascular and mortality risk, making it a robust indicator of systemic dysregulation [13,14,15,16,17]. These diagnostic roles underscore the need for a deeper understanding of its molecular dynamics and substrate interactions. However, despite decades of biochemical and structural studies characterizing ALP’s activity, including its enzymatic conversion of p-nitrophenyl phosphate (PNPP) to p-nitrophenol (PNP), crystallographic snapshots of catalytic intermediates, and kinetic analysis under different pH and metal ion conditions [1,14,15,18,19,20,21,22], details of how ALP interacts with substrates under varying conditions and the resulting molecular dynamics during catalysis remain unclear.

Fourier transform infrared (FTIR) spectroscopy provides a powerful probe of enzymatic transformations, sensitive to bond vibrations in both substrates and protein backbones [23,24,25,26,27,28]. Previous studies have provided structural or kinetic insights, but comprehensive Fourier transform infrared spectroscopy (FTIR)-based analysis of enzymatic processes like ALP activity is limited [29,30,31].

Indeed, most FTIR studies to date have primarily focused on the mid-infrared (mid-IR) region (typically 900–1800 cm^−1^), which contains the characteristic amide I and II bands. The amide I band arises mainly from C=O stretching, and the amide II band originates from N–H bending coupled to C–N stretching; together, these modes inform on protein secondary structure and the fingerprint vibrations associated with substrate and product functional groups.

In contrast, the low-frequency region (500–900 cm^−1^), which reports on metal–ligand and phosphate-related modes, and the high-frequency range (2800–4000 cm^−1^), which includes C–H, N–H, and O–H stretching modes, sensitive to hydration and hydrogen bonding, remain largely unexplored in ALP.

In this paper, we utilize state-of-the-art FTIR spectroscopy to monitor in situ the real-time dynamics of the catalytic conversion of PNPP to PNP by probing the full vibrational spectra, from 500 to 4000 cm^−1^. To avoid misunderstanding, we note that the objective of this study is to monitor the vibrational evolution of substrates and products during catalysis using full-range FTIR, rather than to benchmark reaction efficiency or replace conventional kinetic assays. Accordingly, this approach is intended to provide mechanistic insight into catalytic processes, not quantitative evaluation of catalytic performance. By capturing both conventional mid-IR structural markers and underexplored low- and high-frequency features, a more holistic view of the molecular environment, protein secondary structure, metal–center interactions, and catalytic function of ALP can be obtained. This approach complements kinetic and crystallographic studies by providing a continuous spectroscopic fingerprint of catalysis under near-native aqueous conditions.

## 2. Materials and Methods

All experiments were performed under controlled aqueous conditions using full-range attenuated total reflection FTIR on a Nicolet iS50 spectrometer (Thermo Scientific, Waltham, MA, USA) equipped with a diamond ATR accessory, to monitor ALP-catalyzed PNPP hydrolysis in real time. 

### 2.1. Materials

Alkaline phosphatase (ALP) and its corresponding assay buffer were prepared and provided by the Bustamante laboratory (Berkeley, CA, USA). The ALP sample was originally expressed and purified following the procedures described in Chen et al., PNAS 2020 (117, 24740–24748). *E. coli* alkaline phosphatase (PhoA) was purified under native conditions and supplied in buffer containing 20 mM Tris–HCl (pH 8.0) and 100 mM NaCl, without added divalent cations. The enzyme stock was stored on ice and diluted to the desired concentration immediately before each measurement. All other chemicals, including p-nitrophenyl phosphate (PNPP, Cat. No. 34045), p-nitrophenol (PNP, Cat. No. 1048), and inorganic phosphate (Pi, Cat. No. 10385405), were purchased from Thermo Scientific (Waltham, MA, USA) and used without further purification. All aqueous solutions were prepared using ultrapure water (18.2 MΩ·cm) and the same 20 mM Tris·HCl (pH 8.0) and 100 mM NaCl.

### 2.2. Sample Preparation

For the static FTIR experiments, four reference solutions were prepared: 20 µM ALP, 610 mM PNPP, 115 mM PNP, and 183 mM Pi. All solutions were dissolved in the same buffer described above. The concentrations of ALP, PNPP, PNP, and Pi were individually adjusted to optimize the FTIR signal for each species, yielding comparable peak intensities while accounting for their distinct infrared absorption cross-sections, solubility limits, and molecular sizes.

For the time-resolved (reaction) experiments, ALP and PNPP stock solutions (20 µM and 610 mM, respectively) were mixed directly on the ATR crystal in equal volumes (5 µL + 5 µL) of the two solutions to yield a 10 µL reaction mixture containing 10 µM ALP and 305 mM PNPP. This configuration allowed us to initiate and monitor the enzymatic hydrolysis of PNPP in situ, providing a continuous spectral record of product formation and substrate consumption. Additional mixtures were prepared by varying the ALP and PNPP concentrations to obtain (i) a high-ALP condition (10 µM ALP, 61 mM PNPP) and (ii) a low-ALP condition (1 µM ALP, 305 mM PNPP). All solutions were freshly prepared in the same buffer immediately before each measurement and maintained at PH 8. The pH remained stable during mixing and reaction, as the added enzyme and substrates had negligible buffering capacity compared with the Tris buffer used throughout this study.

### 2.3. FTIR Measurements

All FTIR measurements were performed at 295 ± 2 K using a Nicolet iS50 spectrometer (Thermo Scientific, Waltham, MA, USA) equipped with a diamond ATR accessory. Figure 1a illustrates the ATR-FTIR optical layout and sampling geometry, together with the data processing sequence from interferogram to absorbance spectra. In the Michelson interferometer, a KBr beamsplitter splits the broadband IR beam into a fixed-mirror arm and a moving-mirror arm; their recombination produces a multiplexed interferogram that is Fourier transformed into a spectrum. In ATR, the beam undergoes total internal reflection in the diamond, generating an evanescent field that samples approximately 1–2 µm of the droplet before reaching the DTGS detector.

The spectrometer was routinely wavenumber-calibrated with a standard polystyrene film to ensure spectral accuracy. Spectra were collected over 400–4000 cm^−1^ at a resolution of 4 cm^−1^. Data spacing was 0.25 cm^−1^ for static spectra and 0.5 cm^−1^ for time-resolved acquisitions. Prior to each measurement, an instrument background spectrum was recorded with a clean, dry ATR crystal to calibrate the detector response. A buffer background was collected at the beginning of each session (one for the static data day and one for the time-resolved day), and the corresponding session background was used for that session’s measurements. Approximately 10 µL of sample was deposited on the ATR crystal for each static measurement.

For the static spectra, 256 scans were averaged per acquisition to maximize signal-to-noise. The ATR micro-well was left open (no lid), so the droplet was exposed to ambient air.

For the time-resolved experiments, 64 scans were averaged per spectrum to balance acquisition speed and spectral quality, yielding < 1 min per spectrum and a temporal resolution of 1.5 min. The reaction was initiated by manually mixing ALP and PNPP solutions directly on the ATR crystal for approximately 10 s. The ATR micro-cell was then sealed to minimize evaporation, using the iS50 ATR liquid holder volatile cover (Thermo Scientific, P/N 470-470300). Time zero (t = 0) was defined as the start of acquisition.

All measurements were performed at constant pH 8.0 ± 0.1, corresponding to the optimal catalytic condition of ALP. Spectra were acquired using OMNIC software (version 9; Thermo Scientific) and further processed in Python (v3.10.18) for subsequent analysis and visualization.

### 2.4. Data Analysis

All FTIR spectra were processed and analyzed using custom in-house Python scripts. Raw absorbance spectra were first corrected for baseline drift and buffer contribution by subtracting the corresponding blank spectrum measured under identical conditions.

Quantitative analysis was performed by fitting predefined spectral windows with standard Voigt line-shape model (Gaussian σ and Lorentzian γ) plus a linear baseline. Voigt line shapes were selected because solution-phase IR bands typically reflect a convolution of homogeneous broadening (lifetime and interaction effects) and inhomogeneous broadening (environmental distributions). In our data, pure Gaussian or pure Lorentzian profiles produced higher residuals and poorer AIC/BIC values, whereas the Voigt model provided a more accurate and physically consistent representation of the line shapes. A single-peak Voigt model was used by default. Two- or three-Voigt component fits were applied only where derivative inspection or residual analysis indicated partial overlapping components. Initial parameters were estimated from lightly smoothed spectra using a Savitzky–Golay filter (2nd order, 5–10 points) to assist derivative-based peak detection; this step was used solely for initialization and not for quantitative fitting or presentation.

For static reference spectra (ALP, PNPP, PNP, and Pi), peak centers were freely fitted within broad physically reasonable ranges (typically ±20–30 cm^−1^ from literature values). These fitted positions were then used as reference values to constrain the corresponding peaks in the time-resolved series (±5–10 cm^−1^). All fits were performed using bounded non-linear least squares with non-negative amplitudes, center positions constrained with ±5–10 cm^−1^ of the reference values from static spectra, and Gaussian/Lorentzian widths limited to physically reasonable ranges (σ = 1–15 cm^−1^, γ = 1–20 cm^−1^) to prevent overfitting. Fit quality was verified by residual analysis, and fits showing systematic deviations were manually reviewed.

For time-resolved FTIR measurements, each time slice of the reaction series was analyzed independently following the same baseline correction and windowed Voigt-fitting workflow described above. The resulting spectra were then compared through direct spectral tracking and derivative-based differential analysis, as detailed in the Results Section.

## 3. Results

### 3.1. Static Absorbance of ALP, PNPP, PNP and Pi

The first step to understanding the real-time dynamics of the catalytic conversion is to study the full vibrational spectra, in static conditions, of each of its constituents.

Figure 1b–e report the static FTIR–ATR spectra of ALP, PNPP, PNP, and Pi. The spectra can be divided into five characteristic regions, including X–H vibrations, aromatic/C=O vibrations, amide modes, phosphate modes, and skeletal region.

Panel b shows the FTIR–ATR spectrum of ALP (20 µM, pH 8 buffer, RT), revealing the typical broad and complex vibrational pattern of a folded protein, characterized by a broad band in the high-frequency region (X–H stretching), strong amide bands between 1700 and 1500 cm^−1^, and multiple modes in the fingerprint range (1200–900 cm^−1^) arising from protein side-chain and backbone vibrations. These features reflect contributions from backbone modes (Amide A, I, II, III), aliphatic C–H stretches, and phosphate-associated bonds, characteristics of glycoproteins.

By comparing observed wavenumbers to standard infrared reference tables [32,33,34,35], protein spectroscopy reviews [23,36,37], and published FTIR analyses of amide and phosphate vibrations [23,35,38], we can assign the following modes to each peak in the spectra.

In the high-frequency X–H region (4000–2500 cm^−1^), the broad peak near 3398 cm^−1^ corresponds to Amide A (N–H stretching), sensitive to hydrogen bonding within the protein backbone [23,36,37], while the two weaker features at 2944 and 2889 cm^−1^ are assigned to alkyl C–H stretches from protein side chains [32,39].

In the Amide I region between 1700 and 1620 cm^−1^, we distinguish multiple peaks. The peak near 1708 cm^−1^ is attributed to β-turn structures (C=O stretching, hydrogen-bonded) [23,40], while the peak at 1624 cm^−1^ arises from intermolecular β-sheet arrangements [23,36]. These Amide I peaks primarily reflect C=O stretching coupled to CN and NH modes and provide insights into secondary structure.

In the Amide II region, the peaks at 1551 and 1531 cm^−1^ correspond to the Amide II band, dominated by N–H in-plane bending and C–N stretching, with contributions from C=O bending [23,36,38]. Although these peaks are mainly associated with the peptide bond vibrations, they also provide information on hydrogen bonding.

In the mid-frequency side-chain region (1500–1200 cm^−1^), the peak at 1459 cm^−1^ is assigned to CH_2_ scissoring vibrations [32,41], and the nearby peak at 1418 cm^−1^ to symmetric COO^−^ stretching, indicative of carboxylate groups in acidic residues [42]. A weaker mode at 1334 cm^−1^ can be resolved and is associated with the CH_2_ wagging vibrations [32,43].

The strongest contribution to the FTIR spectra is observed in the amide III region (1300–1200 cm^−1^), where the strong peak at 1215 cm^−1^ arises from coupled N–H bending and C–N stretching vibrations. This band is strongly hydrogen-bond sensitive, and its position is directly related to the backbone conformation [23,36,37].

Phosphate/carbohydrate region (1200–900 cm^−1^): Several strong peaks appear in the 1100–900 cm^−1^ range. The band at 1112 cm^−1^ is consistent with C–O stretching and carbohydrate-associated vibrations [35]. A peak at 1043 cm^−1^ reflects carbohydrate C–O/C–O–C contributions [35]. The band at 994 cm^−1^ corresponds to mixed protein fingerprint modes, including C–N and C–C skeletal vibrations [33]. Finally, the 923 cm^−1^ band is attributed to protein skeletal or carbohydrate vibrations [44]. Overall, since neither the buffer nor the enzyme contains significant inorganic phosphate, the peaks in this region arise from protein and glycan vibrations rather than phosphate species.

This complexity reflects the enzyme’s glycoprotein nature, its secondary structure content, and the importance of phosphate interactions in catalysis.

A consolidated list of observed bands and their assignments is provided in Table 1.

Panels (c–e) show the FTIR spectra of PNPP, PNP and PI. Contrarily to ALP, which reveals broad amide bands (Amide I and II) between 1500 and 1700 cm^−1^ and complex multiplets in the fingerprint region, PNPP, PNP, and Pi exhibit narrow peaks concentrated mainly between 1800 and 900 cm^−1^, consistent with their reduced chemical complexity. In the high-frequency region (4000–2800 cm^−1^), only PNP reveals a broad O–H band, while PNPP and Pi remain largely featureless, providing a distinct spectral handle for identifying the reaction product.

All peak identifications were derived from experimental spectral databases [45,46,47] and from established group-frequency compilations [35,48,49].

Panel c shows the spectrum of PNP, which reveals how, after hydrolysis, the product retains aromatic ring modes, with C=C stretching near 1600 cm^−1^ and a broad O–H band at 3200–3600 cm^−1^, reflecting the phenolic hydroxyl group [49,50]. The persistence of NO_2_ peaks (1320 and 1390 cm^−1^), albeit shifted, highlights change in the electronic environment due to the removal of the phosphate group [45,48]. These spectral differences between PNPP and PNP provide a molecular fingerprint of the enzymatic reaction outcome. The combined appearance of the O–H band and slight downshift of the NO_2_ doublet serve as direct evidence of phosphate cleavage and phenolic product formation.

Panel d shows the spectrum of PNPP, which displays strong nitro group vibrations at 1496 and 1508 cm^−1^, corresponding to symmetric and asymmetric NO_2_ stretching. These peaks are diagnostic of aromatic nitro compounds and are sensitive to the electronic environment [45,48,49]. Additionally, phosphate-related P–O vibrations appear around 850–1100 cm^−1^, consistent with the presence of a phosphate ester group [35]. These bands are particularly important because they diminish during ALP-catalyzed hydrolysis, serving as spectral markers of substrate consumption. The phosphate ester vibrations are vividly distinct from the aromatic and nitro regions, allowing unambiguous tracking of PNPP depletion during the catalysis reaction.

The Pi spectra is shown on panel e. The Pi spectrum exhibits strong, sharp bands between 900 and 1200 cm^−1^, corresponding to symmetric and asymmetric P–O and P=O stretching vibrations [34,44,51,52]. These modes serve as clear markers of phosphate release during ALP catalysis. A broad O–H stretching envelope (3000–3700 cm^−1^) and a H_2_O bending mode near 1655 cm^−1^ reveal hydration of the H_2_PO_4_^−^/HPO_4_^2−^ species [49]. The relative simplicity of the Pi spectrum, compared to PNPP and PNP, makes it a reliable indicator of reaction completion and product accumulation. Compared to PNPP and PNP, the Pi spectrum lacks aromatic bands entirely and has sharp phosphate vibrations, making it a clean reference for observing free phosphate accumulation.

### 3.2. In Situ Spectra of PNPP Reaction Under High ALP

Upon mixing alkaline phosphatase with p-nitrophenyl phosphate, time-resolved Fourier transform infrared spectra were collected to monitor the reaction progress (Figure 2). The absorbance map in Figure 2b summarizes the full-time evolution (red = higher absorbance, blue = lower), with time zero defined by the onset of mixing and sealing. Selected bands increase or decrease depending on whether they report product formation or substrate depletion.

In the aromatic/nitro window (1700–1400 cm^−1^), two reproducible changes are resolved in the time cuts (Figure 2d). First, within 1600–1620 cm^−1^, the feature that visually appears to drift is explained by intensity redistribution between two overlapping components near ~1606 and ~1617 cm^−1^: the p-nitrophenyl-phosphate-related mode (~1606 cm^−1^) decreases while the p-nitrophenol-related mode (~1617 cm^−1^) grows, with fitted centers remaining essentially constant within uncertainty. Second, the envelope near ~1592 cm^−1^ resolves into a doublet at ~1595 and ~1583 cm^−1^ that remains visible at the latest time cut (t_4_). In addition, small blue shifts are observed for the ~1510 → ~1518 cm^−1^ and ~1494 → ~1499 cm^−1^ components. In the static references, p-nitrophenyl phosphate (~1591 cm^−1^) and p-nitrophenol (~1595 cm^−1^) delimit this aromatic C=C region, whereas alkaline phosphatase and inorganic phosphate show no isolated features there; the 1510/1495 cm^−1^ pair lies in the nitro window of the substrate/product with only limited influence from the alkaline phosphatase amide II envelope (Section 3.1).

In the fingerprint range, the trends align with the inorganic-phosphate and p-nitrophenyl-phosphate references (Figure 2e,f). The 1077 cm^−1^ band increases monotonically toward a plateau, matching the inorganic-phosphate marker identified in Section 3.1; the ~990 cm^−1^ region is not used as a primary readout because it overlaps with the alkaline phosphatase ~925 cm^−1^ band and the long tail of inorganic phosphate ~847 cm^−1^. Within 1400–1200 cm^−1^, the ~1345 → ~1340 cm^−1^ and ~1294 → ~1290 cm^−1^ bands exhibit small red shifts that reside in the substrate-side window, while alkaline phosphatase contributes only weak CH_2_/carbohydrate features around ~1334 cm^−1^ and inorganic phosphate has no strong modes there. As a stability check, the 1460 cm^−1^ CH_2_ scissoring band (present in alkaline phosphatase and p-nitrophenol) shows negligible center motion.

Quantified trajectories for the four tracked centers are shown in Figure 2g–j, confirming two blue shifts (1510 → 1518 cm^−1^; 1494 → 1499 cm^−1^) and two red shifts (1345 → 1340 cm^−1^; 1294 → 1290 cm^−1^) that evolve smoothly toward steady values under the high-concentration condition; fitting and normalization procedures are described in Methods.

### 3.3. ALP Concentration Dependence

We also quantified peak centers over time at low, intermediate, and high [ALP] (Methods). Figure 3 assembles these datasets to highlight how the structural response scales with enzyme loading. The time–wavenumber maps for the three concentrations (Figure 3a–c) show similar overall evolution in the 1700–900 cm^−1^ region, but the amplitude of the changes in the aromatic and fingerprint windows clearly decreases from high to mid to low [ALP]. Normalized spectra taken at t = 0 and 33 min in two structural windows (1250–1350 and 1050–1150 cm^−1^; Figure 3d–g) make this trend more explicit. At t = 0 min, all three traces share the same set of bands and very similar peak positions, with only modest intensity differences that reflect the varying amount of ALP in the mixture. By t = 33 min, however, the high-[ALP] spectra show the strongest reshaping: in the 1250–1350 cm^−1^ window (Figure 3e) the two PNPP fingerprint features near 1345 and 1294 cm^−1^ are strongly redistributed and slightly shifted, whereas in the 1050–1150 cm^−1^ window (Figure 3g) the emerging inorganic-phosphate band near 1077 cm^−1^ produces a pronounced low-frequency shoulder on the PNP band. The mid-[ALP] spectra follow the same qualitative pattern with reduced magnitude, and the low-[ALP] spectra remain much closer to their t = 0 min profiles, with only a modest Pi shoulder and limited distortion of the fingerprint envelope.

These qualitative trends are captured more compactly by the peak center analysis in Figure 3h,i. At the initial time, all four markers have the same centers within our resolution—approximately 1510, 1494, 1345, and 1294–1295 cm^−1^—so the dashed gray lines connecting the t = 0 min points are essentially flat across [ALP]. Subsequent motion, however, scales strongly with concentration. In the nitro/aromatic window, the band near 1510 cm^−1^—assigned to a mode in the NO_2_/aryl C=C region—blue shifts to 1518 cm^−1^ at high enzyme concentration (Δν = +8 cm^−1^), while at mid concentration it moves only to 1510.542 cm^−1^ (Δν = +0.80 cm^−1^) and at low concentration it is unchanged within resolution (1509.4 → 1509.4 cm^−1^). Its companion near 1494 cm^−1^—also within the NO_2_-coupled aromatic manifold—blue shifts to 1499 cm^−1^ at high concentration (Δν = +5 cm^−1^), but only +0.30 cm^−1^ at mid concentration (1493.9 → 1494.2 cm^−1^) and +0.18 cm^−1^ at low concentration. In the fingerprint-I window, the ~1345 cm^−1^ marker—dominated by sym ν(NO_2_) with contributions from phenolic ν(C–O)—red shifts to 1340 cm^−1^ at high concentration (Δν = −5 cm^−1^), shows a smaller change at mid concentration (1344.8 → 1343.1 cm^−1^; Δν = −1.7 cm^−1^), and exhibits no resolvable shift at low concentration (1345 → 1345 cm^−1^). The ~1294–1295 cm^−1^ marker—assigned to a phenolic C–O/substrate-side fingerprint mode—red shifts to 1290 cm^−1^ at high concentration (Δν = −4 cm^−1^), to 1290.7 cm^−1^ at mid concentration (Δν = −3.8 cm^−1^), and changes only slightly at low concentration (1295 → 1294 cm^−1^; Δν = −1 cm^−1^). Together, the spectral slices in Figure 3d–g and the peak center summary in Figure 3h,i show that two nitro/aromatic modes blue shift and two substrate-side fingerprint modes red shift, with shift amplitudes that diminish systematically from high to mid to low enzyme concentration, while all four centers share common starting positions across the series.

## 4. Discussion

Static spectra of alkaline phosphatase, p-nitrophenyl phosphate, p-nitrophenol, and inorganic phosphate provide the reference framework for three robust time-resolved outcomes in our in-situ series: clean product growth at 1077 cm^−1^, a structured evolution in the ~1590–1620 cm^−1^ aromatic window, and small red shifts in the 1400–1200 cm^−1^ fingerprint region. The 1077 cm^−1^ band is strong and comparatively isolated in the phosphate spectrum, whereas the ~990 cm^−1^ phosphate band lies in a region lifted by the protein (~923–925 cm^−1^) and the broad phosphate feature near ~847 cm^−1^. Consistent with this, the 1077 cm^−1^ trajectory rises monotonically toward a plateau across enzyme concentrations and serves as the primary product readout (Figure 2; Methods). We emphasize that these measurements provide a structurally resolved spectroscopic view of catalytic progression and are not intended to extract quantitative kinetic parameters; traditional assays such as UV–Vis or HPLC remain the standards for determining enzymatic activity, and FTIR here serves as a complementary structural probe [27,53].

Although these spectral trajectories impose valuable structural constraints on the catalytic process, they do not, by themselves, establish a complete mechanistic pathway. Resolving detailed chemical mechanisms would require complementary approaches, such as isotope labeling, mutagenesis, intermediate trapping, or time-resolved structural methods, which lie beyond the scope of the present work.

In the aromatic window under high enzyme loading, two reproducible observations emerge. The envelope near ~1592 cm^−1^ resolves at late times into a ~1595/1583 cm^−1^ doublet, while the feature that visually appears to drift within 1600–1620 cm^−1^ is accounted for by time-dependent intensity redistribution between two overlapping components near ~1606 and ~1617 cm^−1^; within fit uncertainty, the component centers themselves remain essentially stationary. Static references place p-nitrophenyl phosphate near ~1591 cm^−1^ and p-nitrophenol near ~1595 cm^−1^, so the resolved doublet may reflect the coexistence of substrate-like and product-like environments; a minor phenolate-like contribution near 1595 cm^−1^ is compatible with product buildup and small adjustments in aromatic conjugation. Alternatively, local binding or electrostatic heterogeneity could produce minor vibrational splitting without chemical change. Small blue shifts of the ~1510 → ~1518 cm^−1^ and ~1494 → ~1499 cm^−1^ features fall within the nitro/aromatic manifold and likely report modest adjustments in local electrostatics, hydrogen-bonding networks, or ionization equilibria rather than wholesale changes in normal-mode identity.

In the fingerprint-I window, the ~1345 → ~1340 cm^−1^ and ~1294 → ~1290 cm^−1^ markers exhibit small but consistent red shifts under high enzyme loading. Because these bands sit in the substrate-side fingerprint region, their evolution can arise from composition-driven centroid changes as substrate is replaced by product, from gradual reorganization of solvation and hydrogen-bond networks around the nitro/phenoxide system, and from weak protein-side contributions around ~1334 cm^−1^. Such red shifts are in line with minor relaxation of C–O and P–O bonds after PNPP cleavage and with the increased charge delocalization linked with phenolate accumulation. We therefore use these bands primarily as structural reporters rather than as the principal quantitative channel.

Beyond the vibrational assignments discussed above, it is also useful to consider how full-range FTIR compares with other commonly used spectroscopic techniques. In the broader spectroscopic context, full-range ATR-FTIR provides information that is complementary to Raman, luminescence-based assays, and NMR spectroscopy. Raman spectroscopy offers sensitivity to symmetric and nonpolar vibrational modes and is valuable for probing aromatic ring and backbone motions; however, Raman intensities for nitro and phosphate functional groups are comparatively weak, limiting its ability to follow PNPP→PNP conversion with the same specificity achieved here [23,54]. Luminescence- and UV–Vis-based assays provide highly quantitative kinetic measurements of enzymatic turnover but do not directly report on the molecular-level structural evolution of local bonding environments or structural microstates [55,56]. NMR spectroscopy yields atomic-level structural information but is less suited for temporal resolution and vibrational specificity required to follow continuous catalytic progression in real time [57]. Full-range FTIR therefore occupies a distinct niche by simultaneously capturing substrate, product, and protein-associated modes across the entire vibrational manifold, offering a structurally resolved and time-continuous view of catalysis that complements, rather than replaces, these other modalities.

A consistent concentration dependence is observed when comparing high, mid, and low enzyme loadings (Figure 3). The initial peak centers are indistinguishable within resolution across conditions, indicating that the starting vibrational landscape is independent of enzyme concentration. By contrast, the amplitudes of the subsequent shifts decrease monotonically with decreasing enzyme concentration and, at the lowest loading, fall below detectability. This behavior can be rationalized without invoking a single specific mechanism: deeper conversion within the fixed 33 min acquisition window produces larger composition-weighted centroid changes in overlapping bands; higher site occupancy at high loading amplifies local field and hydrogen-bond perturbations associated with binding; slower apparent kinetics at low loading reduce the net change observable within the time window; and the combination of signal-to-noise and residual drift compensation imposes a finite detection threshold for small shifts (0–8 cm^−1^).

To further enhance the monitoring capability of the approach, several experimental and analytical extensions may be incorporated in future studies. Higher temporal resolution—via rapid-scan acquisition or pump–probe modalities—would allow more detailed observation of transient spectral changes and better separation of overlapping vibrational components. Systematic variation in solution conditions such as pH, ionic strength, or solvent isotope composition could help distinguish compositional effects from microenvironmental rearrangements. In addition, polarization- or orientation-sensitive ATR measurements would provide access to mode-specific dipole reorientation or binding geometry, while global kinetic fitting or multivariate decomposition could complement windowed peak tracking by more robustly isolating subtle spectral contributions.

Several experimental limitations should be considered when interpreting the present FTIR data. Water-background absorption and associated baseline drift impose constraints on sensitivity, particularly near strong solvent-related features, while spectral congestion from overlapping protein- and substrate-derived vibrations limits unique peak assignment in certain fingerprint regions. In addition, finite signal-to-noise ratios define a practical detection threshold for small peak shifts, especially at low enzyme loading [23,58,59,60].

Controls and limitations were implemented to avoid conflating thickness or carrier effects with chemical shifts. The 1460 cm^−1^ CH_2_ scissoring band, present in protein and p-nitrophenol, is thickness-sensitive and is therefore used as a stability check rather than for normalization. The protein-associated band near ~1045 cm^−1^ remains comparatively stable in center and width, and the ~925 cm^−1^ band is robust at high and mid enzyme loadings but often falls below detectability at low loading. For the ~1590–1620 cm^−1^ window, the spectral bounds and time evolution reported here capture the dominant experimental trends. While the observed behavior is consistent with an intensity redistribution scenario, alternative interpretations such as binding-mode heterogeneity or coupling-induced line-shape changes cannot be fully excluded.

While the present study focuses on a well-controlled enzyme–substrate system, the general approach may be extended to more complex biological or heterogeneous matrices with appropriate experimental considerations. In such environments, additional challenges are expected, including increased scattering, stronger water-pathlength effects, and more congested background signals. These issues would require careful control of sampling geometry, background subtraction, and spectral referencing. In this context, ATR configurations remain advantageous due to their shallow penetration depth, and integration with microfluidic flow cells or surface-confined geometries could improve reproducibility and background control. Selective isotopic labeling or targeted spectral windows may further aid in isolating reaction-specific vibrational signatures from complex backgrounds [24,58,59]. These considerations suggest that full-range FTIR is adaptable to more complex systems, while emphasizing the need for tailored experimental design.

Taken together, the static-anchored analysis yields a compact and reproducible infrared readout of turnover and structural evolution: 1077 cm^−1^ tracks product formation; the ~1590–1620 cm^−1^ region reports a structured development that includes late-time doublet resolution and intensity redistribution between overlapping modes; and 1345/1294 cm^−1^ captures small substrate-side red shifts. The amplitudes of these changes diminish systematically with decreasing enzyme concentration, whereas the starting frequencies are common across conditions. Future experiments that vary ionic strength and pH, implement solvent isotope substitution, or use competition/site-blocking assays—ideally combined with polarization- or orientation-sensitive ATR measurements—should help to disentangle composition changes from microenvironment and binding effects within the same static–dynamic framework.

## 5. Conclusions

This study demonstrates the capability of full-range, time-resolved FTIR spectroscopy to monitor in situ the catalytic conversion of PNPP to PNP by ALP across the entire 500–4000 cm^−1^ range. Static spectra of the pure components, ALP, PNPP, PNP, and PI, provided the essential vibrational framework that enabled clear differentiation of substrate, product, and enzyme contributions. By integrating static references and time-resolved tracking, this workflow yields a compact yet information-rich readout of enzymatic turnover.

These references anchored quantitative tracking of spectral evolution during catalysis and ensured robust deconvolution of overlapping modes.

In the reaction series, the 1077 cm^−1^ phosphate stretch emerged as a reliable marker of product formation, rising monotonically to a plateau under all enzyme concentrations. The aromatic region (1580–1620 cm^−1^) revealed a structured evolution consistent with substrate-to-product transition: a doublet appeared near 1583/1595 cm^−1^ and a smaller shift in the nitro band near 1510 cm^−1^. These trends capture subtle electronic and bonding rearrangements of the p-nitrophenyl group accompanying hydrolysis. Meanwhile, small but reproducible red shifts in the PNPP fingerprint-I window (1294 → 1290 cm^−1^, 1345→1340 cm^−1^) reflected progressive substrate depletion and possible weak coupling to enzyme-side vibrations.

Beyond establishing a framework for ALP, full-range FTIR measurements offer a broadly generalizable strategy to monitor the time evolution of substrate- and product-specific vibrational bands over time, providing quantitative information on molecular bonding and qualitative insight into catalytic progression and associated molecular-level structural changes during turnover.

## Figures and Tables

**Figure 1 biology-15-00068-f001:**
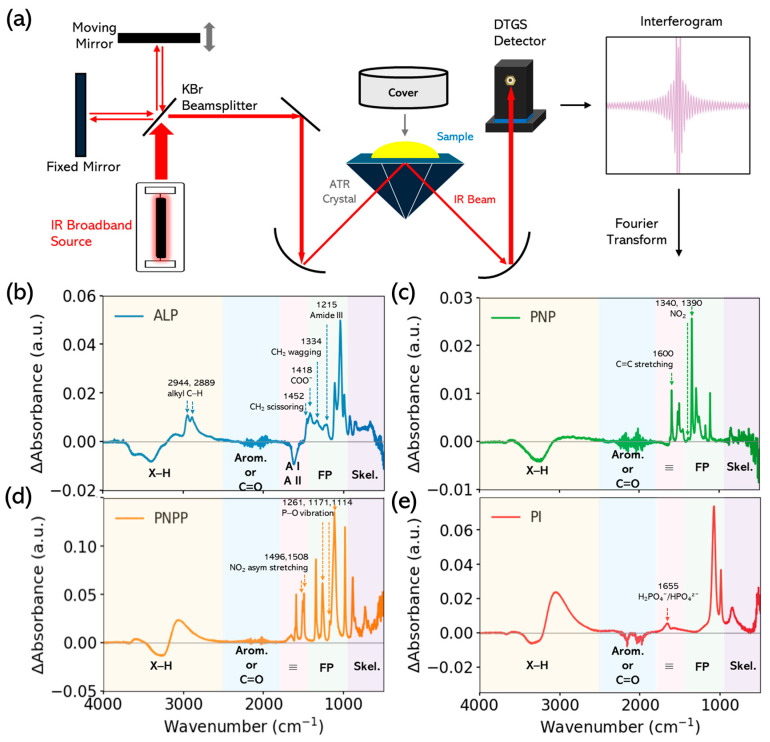
ATR-FTIR layout and reference spectra. (**a**) A broadband SiC (“Globar”) source is split by a KBr beamsplitter into the fixed- and moving-mirror arms of a Michelson interferometer; recombination produces a time-dependent interferogram that is Fourier transformed to yield spectra. During measurement, the droplet on the diamond ATR was covered to limit evaporation and atmospheric H_2_O/CO_2_ pickup. Total internal reflection generates an evanescent field that probes ~1–2 µm of the sample before detection by the DTGS (KBr) detector. (**b**–**e**) Reference ATR-FTIR spectra of the individual components: (**b**) ALP (20 µM; pH 8 buffer), (**c**) PNP (115 mM), (**d**) PNPP (610 mM), (**e**) Pi (183 mM). Spectra were recorded on a Nicolet iS50 (diamond ATR, KBr beamsplitter, DTGS detector) over 500–4000 cm^−1^ at 4 cm^−1^ resolution; air served as the background, and the buffer spectrum was subtracted.

**Figure 2 biology-15-00068-f002:**
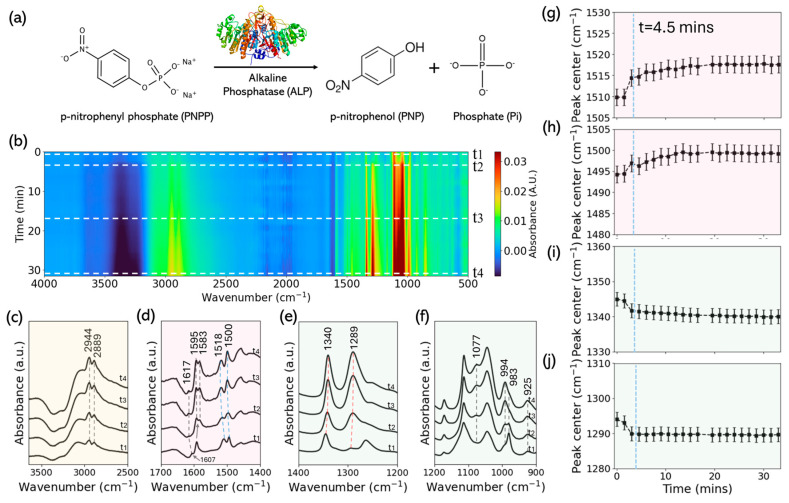
ALP-catalyzed PNPP hydrolysis monitored by ATR-FTIR (high [ALP] dataset). (**a**) Reaction scheme. (**b**) In situ absorbance map (500–4000 cm^−1^) with four time cuts marked by horizontal dashed lines (t_1_–t_4_ = 0, 4.5, 25.5, 33 min); dashed rectangles indicate analysis windows. (**c**–**f**) Spectral snapshots at t_1_–t_4_ from the same run in three windows: (**c**) C–H stretching (≈3500–2500 cm^−1^); (**d**) nitro/aromatic region (1700–1400 cm^−1^) highlighting the emergence of a doublet near ~1592 cm^−1^ that resolves into 1595 and 1583 cm^−1^ at late times, together with blue-shifting bands at 1510 → 1518 cm^−1^ and 1494 → 1499 cm^−1^; (**e**) fingerprint-I (1400–1200 cm^−1^) showing red shifts 1345 → 1340 cm^−1^ and 1294 → 1290 cm^−1^; (**f**) fingerprint-II/phosphate window (1115–900 cm^−1^) showing monotonic growth of the inorganic-phosphate band near 1077 cm^−1^. Spectra are vertically offset for visual clarity; *y*-axis values are shown in arbitrary units. (**g**–**j**) Peak center trajectories for the four tracked modes corresponding to the windows in (**d**–**e**): (**g**) 1510 → 1518 cm^−1^ (blue shift), (**h**) 1494 → 1499 cm^−1^ (blue shift), (**i**) 1345 → 1340 cm^−1^ (red shift), (**j**) 1294 → 1290 cm^−1^ (red shift). The vertical dashed line at t ≈ 4.5 min in (**g**–**j**) marks the time at which kinetic fits to the 1077 cm^−1^ Pi band indicate that the reaction has effectively reached completion.

**Figure 3 biology-15-00068-f003:**
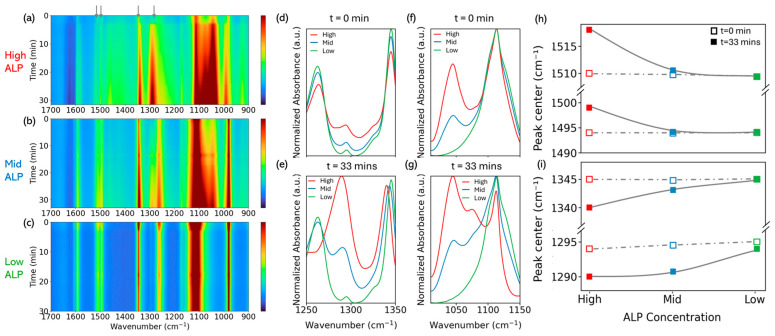
(**a**–**c**) Time–wavenumber ATR-FTIR maps for high, mid, and low ALP concentrations, respectively, showing the evolution of the PNPP/PNP/Pi region over 0–33 min. Gray arrows in (**a**) mark the four bands whose kinetics are analyzed in the subsequent panels. (**d**–**g**) Normalized absorbance spectra at selected windows for the three ALP concentrations (red = High, blue = Mid, green = Low). Panels (**d**,**f**) show the initial spectra at t = 0 min, while panels (**e**,**g**) show the corresponding spectra at t = 33 min, highlighting concentration-dependent band shapes and shifts in the fingerprint regions. (**h**,**i**) Peak center positions extracted from Voigt fits for the aromatic C=C/NO_2_ pair (**h**) and the PNPP fingerprint-I/II pair (**i**) as a function of ALP concentration. Open squares denote initial positions at t = 0 min and filled squares denote final positions at t = 33 min; gray dashed lines connect the t = 0 min peak centers across ALP concentrations and serve as a visual reference for the subsequent shifts.

**Table 1 biology-15-00068-t001:** Peak positions and assignments in the ALP FTIR–ATR spectrum. Observed maxima (cm^−1^), bond assignments, and functional group vibrations are listed.

Peak Position (cm^−1^)	Assignment	Bond Description
3398	Amide A	N–H stretching (hydrogen-bond sensitive)
2944	C–H stretch	Alkyl side chains
2889	C–H stretch	Alkyl side chains
1708	Amide I (β-turn)	C=O stretching + H-bonding
1624	Amide I (β-sheet)	C=O stretching, intermolecular β-sheet
1551	Amide II	NH bending + CN stretching
1531	Amide II	NH bending + CN stretching
1459	CH_2_ scissoring	CH_2_ bending vibrations
1418	COO^−^ symmetric stretch	Carboxylates
1334	CH_2_ wagging	Side-chain CH_2_ vibration
1215	Amide III	NH bending + CN stretching (H-bond sensitive)

## Data Availability

The data presented in this study are available on request from the corresponding author.

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
