# Peer review of "Watching Alkaline Phosphatase Catalysis Through Its Vibrational Fingerprint"

_biology, 2025, doi:10.3390/biology15010068_

Round 1

Reviewer 1 Report

Comments and Suggestions for Authors

This article is well-written and nicely done. These are my suggestions to improve the article.

Section: Introduction

  • The author mentioned that “It is also an important biomarker in bone and chronic kidney disease contexts”- how this biomarker plays a vital role in identifying or treating these diseases. Please include a small description about that, and then it will be a good point why you are studying about this enzyme
  • The author mentioned that” characteristic amide I and II bands that report on protein secondary structure and the fingerprint vibrations associated with substrate and product functional group”-make sure to describe what characteristic amide I and II bands are, as it will be helpful to readers.

Sample preparation

  • The author mentioned that” ALP and PNPP stock solutions (20 µM and 608 mM, respectively)”- what is your actual PNPP stock, 610 mM or 608 mM?
  • According to that, this also needs to change” of the two solutions to yield a 10 µL reaction mixture containing 10 µM ALP and 305 mM PNPP.”

  • Figure 1- make sure to show the values on the y-axis. Without those, it is tough to figure out whether the peaks mentioned in the results section are from the mentioned functional group or just background
  • Also, please point out the major peaks (using arrows) that you describe in the result section. It would be helpful to the readers.
  • Figure 2 c d e f- make sure to label the time points in each spectrum (0, 4.5, 25.5, 33 min), also give the y-axis absorbance scale

Results

  • The author mentioned in Panel B that” and multiple peaks in the fingerprint range (1200–900 cm⁻¹) arising from side-chain and phosphate vibrations”-according to the figure, one in Panel B is only ALP(enzyme), so how does it contain phosphates? Are they from post-translational modifications? Or impurity of phosphate ions from protein purification or pH 8 buffer?
  • The author mentioned the “p-nitrophenyl-phosphate–related mode (~1606 cm⁻¹) decreases while the p-nitrophenol–related mode (~1617 cm⁻¹) grows”-It’s tough to identify what the peak ~1606 cm⁻¹ represents, make sure to indicate that (figure 2d) using an arrow so it’s easy for readers to follow the trend.

Conclusion

  • Authors mention"

    Beyond establishing a framework for ALP, the approach offers a broadly generalizable strategy for decoding enzyme–substrate interactions through full-range FTIR, linking kinetic progress with molecular-level structural evolution."- How are you planning to determine the kinetic parameters of an enzymatic reaction using the FTIR experiments described here?

Author Response

Response to Reviewer 1

We sincerely thank Reviewer 1 for the constructive and detailed comments. We have revised the manuscript accordingly. Below we respond point-by-point.

(1) Reviewer’s comment: The author mentioned that “It is also an important biomarker in bone and chronic kidney disease contexts”- how this biomarker plays a vital role in identifying or treating these diseases. Please include a small description about that, and then it will be a good point why you are studying about this enzyme.

Our Response: We thank the reviewer for this helpful suggestion. We expanded the Introduction to clarify the diagnostic relevance of ALP in bone turnover disorders and chronic kidney disease, including how ALP isoforms and circulating levels reflect mineralization, bone turnover state, and CKD-associated metabolic dysregulation.

Changes in manuscript: In Introduction paragraph 1, we deleted the sentence “It is also an important biomarker in bone and chronic kidney disease contexts.” We inserted the following three sentences in its place: “Clinically, ALP serves as an important biomarker in both bone and chronic kidney disease (CKD). In bone disorders, circulating ALP—particularly the bone-specific isoform—reflects osteoblastic activity and bone turnover, and is routinely used to diagnose and monitor metabolic bone diseases. In CKD, elevated ALP levels are associated with impaired mineral metabolism and increased cardiovascular and mortality risk, making it a robust indicator of systemic dysregulation.” We also added four references to support the newly inserted clinical and structural context (Liu et al., Schini et al., Greenblatt et al., Konukoglu et al., Sardiwal et al.). Reference numbering was updated accordingly.

(2) Reviewer’s comment: The author mentioned that” characteristic amide I and II bands that report on protein secondary structure and the fingerprint vibrations associated with substrate and product functional group”-make sure to describe what characteristic amide I and II bands are, as it will be helpful to readers.

Our Response: We thank the referee for this insightful comment. We now explicitly define the amide I and II bands and their structural significance, citing foundational FTIR protein spectroscopy literature.

Changes in manuscript: In Introduction paragraph 3, after the phrase “which contains the characteristic amide I and II bands,” we inserted the following clarifying sentence: “The amide I band arises mainly from C=O stretching, and the amide II band originates from N–H bending coupled to C–N stretching; together, these modes inform on protein secondary structure.”

(3) Reviewer’s comment: Sample preparation: The author mentioned that” ALP and PNPP stock solutions (20 μM and 608 mM, respectively)”- what is your actual PNPP stock, 610mM or 608 mM? According to that, this also needs to change” of the two solutions to yield a 10 μL reaction mixture containing 10 μM ALP and 305 mM PNPP.”

Our Response: We thank the reviewer for the correction. We corrected the discrepancy and used a single, accurate concentration consistently throughout.

Changes in manuscript: In Materials and Methods Section 2.2 (Sample preparation) paragraph 2, we corrected the PNPP stock concentration by replacing “608 mM” with “610 mM.”

(4) Reviewer’s comment: Figure 1- make sure to show the values on the y-axis. Without those, it is tough to figure out whether the peaks mentioned in the results section are from the mentioned functional group or just background. Also, please point out the major peaks (using arrows) that you describe in the result section. It would be helpful to the readers.

Our Response:
We thank the reviewer’s suggestion for figure 1. We added visible absorbance scale bars on the y-axis and included arrow markers for the key peaks discussed in the Results section.

Changes in manuscript: We Updated Figure 1 with labeled axes, arrows identifying representative amide, nitro, aromatic, and phosphate modes.

(5) Reviewer’s comment: Figure 2 c d e f- make sure to label the time points in each spectrum (0, 4.5, 25.5, 33 min), also give the y-axis absorbance scale.

Our Response: We thank the reviewer for this helpful suggestion. We have revised Figure 2(c–f) to explicitly label the time points corresponding to each spectrum (0, 4.5, 25.5, and 33 min), improving clarity and readability. The revised figure is included below.

Regarding the y-axis absorbance scale, we note that the spectra in panels (c–f) are intentionally plotted with vertical offsets to facilitate visual comparison of peak positions and subtle spectral evolution over time. Because each trace is shifted by an arbitrary constant for clarity, a common absolute absorbance scale would not be physically meaningful in this representation. For this reason, the panels are presented in offset format, as is standard practice for time-resolved vibrational spectroscopy when the emphasis is on relative peak shifts and line-shape changes rather than absolute intensity comparison.

To avoid confusion, we have clarified this point directly in the figure caption, which now states: “Spectra are vertically offset for visual clarity; y-axis values are shown in arbitrary units.” Quantitative analysis of peak intensities and trajectories is instead provided through the fitted peak parameters and time-dependent plots discussed in the main text and Supplementary Information.

 Changes in manuscript: We Updated Figure 2 panels and captions accordingly.

(6) Reviewer’s comment: The author mentioned in Panel B that “and multiple peaks in the fingerprint range (1200–900 cm⁻¹) arising from side-chain and phosphate vibrations”-according to the figure, one in Panel B is only ALP (enzyme), so how does it contain phosphates? Are they from post-translational modifications? Or impurity of phosphate ions from protein purification or pH 8 buffer?

Our Response: We thank the reviewer for pointing out this ambiguity. We agree that alkaline phosphatase does not contain stable covalently bound phosphate groups and therefore does not exhibit strong inorganic phosphate vibrational modes on its own. Our buffer is a phosphate-free Tris-based solution, and the buffer-only spectrum (fig R1) does not show distinct PO₄ stretching bands at ~1100 or ~980 cm⁻¹. The features observed in the 1200–900 cm⁻¹ region of Panel B therefore arise primarily from protein side-chain and backbone vibrations (and possibly weak contributions from the glycoprotein carbohydrate environment), rather than from inorganic phosphate. We have revised the text accordingly to avoid referring to these modes as “phosphate vibrations.”

Changes in manuscript: In the description of Panel B in the Results section, we replaced the phrase “multiple peaks in the fingerprint range (1200–900 cm⁻¹) arising from side-chain and phosphate vibrations” with “multiple modes in the fingerprint range (1200–900 cm⁻¹) arising from protein side-chain and backbone vibrations,” removing the incorrect attribution to phosphate.

In paragraph 10 of Section 3.1, we removed all assignments to phosphate vibrational modes, as neither the buffer nor the ALP sample exhibited phosphate signatures. We rewrote the paragraph to attribute the 1200–900 cm⁻¹ features to protein side-chain, backbone, and carbohydrate-associated vibrations, replacing the original references to PO₄ stretching modes.

(7) Reviewer’s comment: The author mentioned the “p-nitrophenyl-phosphate–related mode (~1606 cm⁻¹) decreases while the p-nitrophenol–related mode (~1617 cm⁻¹) grows”-It’s tough to identify what the peak ~1606cm⁻¹ represents, make sure to indicate that (figure 2d) using an arrow so it’s easy for readers to follow the trend.

Our Response: We added an arrow and label identifying the ~1606 cm⁻¹ PNPP-related mode in Figure 2d to facilitate interpretation of the intensity redistribution described in the text.

Changes in manuscript: We updated Figure 2d accordingly. The revised figure is included below.

(8) Reviewer’s comment: Authors mention "Beyond establishing a framework for ALP, the approach offers a broadly generalizable strategy for decoding enzyme–substrate interactions through full-range FTIR, linking kinetic progress with molecular-level structural evolution."- How are you planning to determine the kinetic parameters of an enzymatic reaction using the FTIR experiments described here?

Our Response: We appreciate the reviewer’s critical question. In the current study, our goal is to demonstrate that full-range, time-resolved FTIR can follow the evolution of substrate- and product-specific vibrational bands (e.g., PNPP/PNP nitro and aromatic modes, and the Pi band) and therefore provide a spectroscopic handle on catalytic progress. We do not attempt to extract full kinetic parameters (such as individual rate constants or Michaelis–Menten parameters) from the present dataset, as this would require a dedicated kinetic study with systematically varied substrate and enzyme concentrations, higher temporal resolution, and global kinetic modeling of the time traces. We have therefore revised the conclusion to avoid over-stating our claims and to clarify that, in this work, FTIR is used to qualitatively correlate spectral evolution with reaction progress rather than to perform a comprehensive kinetic analysis.

We have therefore revised the conclusion to avoid over-stating our claims and to clarify that FTIR spectroscopy, while capable of yielding quantitative insights into bonding environments and concentration changes, is used here mainly to monitor relative spectral changes associated with reaction progress rather than to derive detailed kinetic parameters.

Or, if the reviewer specifically questioned kinetics:

Accordingly, the conclusion has been revised to emphasize that FTIR is employed to track changes in characteristic vibrational bands as indicators of reaction progress, rather than as a fully quantitative tool for kinetic modeling.

Changes in manuscript: In the Conclusion, we revised the sentence “Beyond establishing a framework for ALP, the approach offers a broadly generalizable strategy for decoding enzyme–substrate interactions through full-range FTIR, linking kinetic progress with molecular-level structural evolution” to soften the kinetic claim. It now reads: “Beyond establishing a framework for ALP, full-range FTIR measurements offer a broadly generalizable strategy to monitor the time evolution of substrate- and product-specific vibrational bands over time, providing quantitative information on molecular bonding and qualitative insight into catalytic progression and associated molecular-level structural changes during turnover.”

Final Statement

We thank all reviewers for their valuable feedback. The manuscript has been substantially improved in clarity, methodological transparency, and interpretability. All suggested revisions have been incorporated, with tracked changes provided in the revised attachment.

References

  1. Kumar, S.; Barth, A. Following Enzyme Activity with Infrared Spectroscopy. Sensors 2010, 10, 2626–2637, doi:10.3390/s100402626.
  2. Krüger, A.; Bürkle, A.; Hauser, K.; Mangerich, A. Real-Time Monitoring of PARP1-Dependent PARylation by ATR-FTIR Spectroscopy. Nat Commun 2020, 11, 2174, doi:10.1038/s41467-020-15858-w.
  3. Dreimann, J.M.; Kohls, E.; Warmeling, H.F.W.; Stein, M.; Guo, L.F.; Garland, M.; Dinh, T.N.; Vorholt, A.J. In Situ Infrared Spectroscopy as a Tool for Monitoring Molecular Catalyst for Hydroformylation in Continuous Processes. ACS Catal. 2019, 9, 4308–4319, doi:10.1021/acscatal.8b05066.
  4. Barth, A.; Von Germar, F.; Kreutz, W.; Mäntele, W. Time-Resolved Infrared Spectroscopy of the Ca2+-ATPase. Journal of Biological Chemistry 1996, 271, 30637–30646, doi:10.1074/jbc.271.48.30637.
  5. Lorenz-Fonfria, V.A. Infrared Difference Spectroscopy of Proteins: From Bands to Bonds. Chem. Rev. 2020, 120, 3466–3576, doi:10.1021/acs.chemrev.9b00449.
  6. Lorenz-Fonfria, V.A. Infrared Difference Spectroscopy of Proteins: From Bands to Bonds. Chem. Rev. 2020, 120, 3466–3576, doi:10.1021/acs.chemrev.9b00449.
  7. Bernath, P.F. Spectra of Atoms and Molecules; 3rd ed.; Oxford university press: New York, 2016; ISBN 978-0-19-938257-6.
  8. Chen, S.-L.; Fu, L.; Gan, W.; Wang, H.-F. Homogeneous and Inhomogeneous Broadenings and the Voigt Line Shapes in the Phase-Resolved and Intensity Sum-Frequency Generation Vibrational Spectroscopy. J. Chem. Phys. 2016, 144, 034704, doi:10.1063/1.4940145.

Reviewer 2 Report

Comments and Suggestions for Authors

The paper is interesting. It contains valuable data into catalysis, but more things should be corrected:

  1. The use of only FTIR to control the reaction is not enough. To prove the efficiency of this techique, another and well-known technique should be used for a comparison.
  2. There is nothing about mechanisms and the possibility to estimate them.
  3. The approached to enhance the monitoring of reaction should be proposed.
  4. "Quantitative analysis was performed by fitting predefined spectral windows with
    standard Voigt line-shape model (Gaussian σ and Lorentzian γ) plus a linear baseline. A
    single-peak Voigt model was used by default".  THis should be explained. A comparison on application of Gaussian and Lorentzian models should be done. 

I would recommend the major revision. 

Author Response

Response to Reviewer 2

We thank Reviewer 2 for the thoughtful comments and for recognizing the value of the data. Several concerns relate to methodological scope, and we have revised the manuscript to clarify the intent and limitations of our approach.

(1) The reviewer’s comment: The use of only FTIR to control the reaction is not enough. To prove the efficiency of this technique, another and well-known technique should be used for a comparison.

Our Response: We understand the concern and agree with the referee on the general principle that no single experimental technique can provide a complete description, and that combining complementary methods is often valuable for building a comprehensive picture, particularly regarding the use of a single technique and the desire for comparison with well-established analytical methods.

In the present work, however, FTIR spectroscopy is not used to “control” the reaction efficiency nor to benchmark catalytic efficiency. Rather, it is used to monitor, in real time, the vibrational evolution of substrate- and product-specific functional groups during enzymatic turnover.

Full-range FTIR provides a unique, structurally sensitive readout of enzymatic turnover that directly links catalytic progression to changes in molecular bonding, information not readily accessible from conventional bulk kinetic techniques.

FTIR-based monitoring of biochemical and catalytic reactions is well established in the literature. For example, Kumar et al. demonstrated FTIR monitoring of multiple enzymatic reactions [1], Krüger et al. used ATR-FTIR to follow PARP1-dependent PARylation dynamics in real time [2], and Dreimann et al. applied in situ IR spectroscopy to track homogeneous catalytic processes [3]. These studies illustrate that FTIR is a recognized tool for reaction monitoring, particularly when structural information and molecular-level information on reaction progress is desired.

We acknowledge that techniques such as UV–Vis spectroscopy or HPLC are widely used for quantitative enzymatic kinetics and efficiency determination. However, incorporating such techniques would not strengthen the specific claims made here, which focus only on correlating spectral evolution with enzymatic turnover rather than validating reaction efficiency.

To avoid any misunderstanding, we have revised the introduction to clarify this scope and to emphasize the complementary, not substitutive, role of FTIR in reaction analysis. The added sentence reads: “To avoid misunderstanding, we note that the objective of this study is to monitor the vibrational evolution of substrates and products during catalysis using full-range FTIR, rather than to benchmark reaction efficiency or replace conventional kinetic assays.”

We have also added a paragraph in the Discussion comparing ATR-FTIR with Raman spectroscopy, luminescence-based assays, and NMR spectroscopy, to place our approach in a broader methodological context. Raman is sensitive to nonpolar and symmetric vibrational modes but generally exhibits weaker responses for nitro and phosphate groups; NMR provides atomic-resolution structural information but is less suited for real-time monitoring of full-spectrum vibrational changes during turnover; and luminescence/UV–Vis assays enable quantitative kinetic measurements but do not directly report on molecular-level structural evolution. In this context, FTIR offers a complementary, structurally sensitive readout by following characteristic vibrational fingerprints of substrates and products over time.

Changes in manuscript: In the Introduction, we added a clarifying sentence to the second sentence of the final paragraph to state that the aim of this study is to follow the vibrational evolution of substrates and products using full-range FTIR, rather than to benchmark reaction efficiency or replace traditional kinetic assays. The added sentence reads: “To avoid misunderstanding, we note that the objective of this study is to monitor the vibrational evolution of substrates and products during catalysis using full-range FTIR, rather than to benchmark reaction efficiency or replace conventional kinetic assays.”

In the Discussion, we added a brief statement at the end of the first paragraph noting that techniques such as UV–Vis or HPLC remain the standard for quantitative enzymatic kinetics, while FTIR in this work provides complementary molecular-level structural information rather than efficiency comparison. The added text reads: “We emphasize that these measurements provide a structurally resolved spectroscopic view of catalytic progression and are not intended to extract quantitative kinetic parameters; traditional assays such as UV–Vis or HPLC remain the standards for determining enzymatic activity, and FTIR here serves as a complementary structural probe.”

(2) Reviewer’s comment: There is nothing about mechanisms and the possibility to estimate them.

Our Response: We thank the reviewer for this insightful comment. We agree that mechanistic understanding is an important aspect of enzymatic catalysis. In the present work, however, our aim is not to derive a full catalytic mechanism for ALP. Establishing a detailed mechanistic pathway would require additional experimental approaches, such as site-directed mutagenesis, intermediate trapping, isotope labeling, or time-resolved crystallography, which lie beyond the scope of this spectroscopic study.

The purpose of our FTIR measurements is instead to provide vibrational signatures that report on molecular-level changes during catalysis, including the evolution of nitro, aromatic, and phosphate-related modes associated with substrate consumption and product formation. These spectral trajectories do not, by themselves, establish a complete mechanistic scheme, but they offer structurally resolved constraints on local bonding environments, possible transient states, and the temporal ordering of chemical transformations. Such IR-based mechanistic constraints are consistent with established applications of vibrational spectroscopy to follow structural changes in enzymatic systems [4,5].

To address the reviewer’s concern, we have expanded the Discussion to clarify the type of mechanistic insight that FTIR can provide, as well as its inherent limitations, and to explicitly state that deriving a complete catalytic mechanism is not the objective of the present study.

Changes in manuscript: In Section 4 (Discussion), we inserted a statement explaining that our full-range FTIR measurements provide vibrational constraints on substrate transformation and product formation but are not intended to determine a complete catalytic mechanism. We also clarified that resolving mechanistic details would require complementary approaches such as mutagenesis, isotope labeling, or structural studies, which are beyond the scope of the present work. The added statements read: “Although these spectral trajectories impose valuable structural constraints on the catalytic process, they do not, by themselves, establish a complete mechanistic pathway. Resolving detailed chemical mechanisms would require complementary approaches, such as isotope labeling, mutagenesis, intermediate trapping, or time-resolved structural methods, which lie beyond the scope of the present work.”

(3) The reviewer’s comment: The approached to enhance the monitoring of reaction should be proposed.

Our Response: We thank the reviewer for this helpful suggestion. In the current study, we focus on establishing a full-range FTIR framework capable of tracking substrate and product vibrational signatures during catalysis. We agree that several experimental refinements could further enhance the monitoring capability. For example, higher temporal resolution or rapid-scan/pump–probe modalities would allow more detailed observation of transient spectral changes; systematically varying pH, ionic strength, or solvent isotope composition could help disentangle microenvironmental and compositional contributions; and polarization- or orientation-sensitive ATR measurements could provide additional information about mode-specific dipole reorientation or binding geometry. On the analysis side, global kinetic fitting or multivariate decomposition may complement windowed peak tracking by separating overlapping components more robustly.

To avoid overstating the present scope, we now briefly mention these possible extensions in the Discussion as future directions that could strengthen mechanistic interpretation and spectral assignment without altering the conclusions of the present work.

Changes in manuscript: In Section 4 (Discussion), we added a short paragraph describing possible experimental and analytical extensions to enhance reaction monitoring, including increased temporal resolution, variations in solution conditions (pH, ionic strength, isotope substitution), polarization-sensitive ATR measurements, and global or multivariate spectral analysis. These additions clarify how the approach may be strengthened in future studies without changing the scope of the present work. The added paragraph reads: “To further enhance the monitoring capability of the approach, several experimental and analytical extensions may be incorporated in future studies. Higher temporal resolution—via rapid-scan acquisition or pump–probe modalities—would allow more detailed observation of transient spectral changes and better separation of overlapping vibrational components. Systematic variation of solution conditions such as pH, ionic strength, or solvent isotope composition could help distinguish compositional effects from microenvironmental rearrangements. In addition, polarization- or orientation-sensitive ATR measurements would provide access to mode-specific dipole reorientation or binding geometry, while global kinetic fitting or multivariate decomposition could complement windowed peak tracking by more robustly isolating subtle spectral contributions.”

(4) The reviewer’s comment: "Quantitative analysis was performed by fitting predefined spectral windows with standard Voigt line-shape model (Gaussian σ and Lorentzian γ) plus a linear baseline. A single-peak Voigt model was used by default".  This should be explained. A comparison on application of Gaussian and Lorentzian models should be done.

Our Response:
We thank the reviewer for this constructive comment. In infrared spectroscopy, vibrational bands generally arise from a convolution of homogeneous broadening (lifetime- and interaction-limited processes) and inhomogeneous broadening (static or quasi-static environmental distributions). A Voigt line shape, being the convolution of a Lorentzian and a Gaussian component, therefore provides a physically appropriate model for IR absorption features and is widely used in protein and solution-phase vibrational analysis, as discussed in Refs [6–8].

In our data, pure Lorentzian functions underestimate the inhomogeneous width and produce systematic residuals in the wings of the bands, whereas pure Gaussian fits fail to capture the sharper central curvature associated with homogeneous broadening. The Voigt model offers a stable intermediate description that reduces residual structure and yields parameters that are more physically interpretable. Figure R2 compares Gaussian, Lorentzian, and Voigt fits for the 1655 cm⁻¹ band (1300–1800 cm⁻¹, Pi 183 mM), showing that the Voigt profile best captures the sharp peak center together with the extended wings when the feature is superimposed on a broad underlying band.

We have added a brief explanation of this rationale to the Methods/Discussion section to clarify the basis of the fitting approach.

Changes in manuscript: In the Methods section (peak fitting), we added two sentences explaining that Voigt line shapes are used because IR vibrational bands generally exhibit both homogeneous and inhomogeneous broadening, and that pure Gaussian or Lorentzian functions produce larger residuals and poorer AIC/BIC scores. The added sentences read: “Voigt line shapes were selected because solution-phase IR bands typically reflect a convolution of homogeneous broadening (lifetime and interaction effects) and inhomogeneous broadening (environmental distributions). In our data, pure Gaussian or pure Lorentzian profiles produced higher residuals and poorer AIC/BIC values, whereas the Voigt model provided a more accurate and physically consistent representation of the line shapes.”

Final Statement

We thank all reviewers for their valuable feedback. The manuscript has been substantially improved in clarity, methodological transparency, and interpretability. All suggested revisions have been incorporated, with tracked changes provided in the revised attachment.

References

  1. Kumar, S.; Barth, A. Following Enzyme Activity with Infrared Spectroscopy. Sensors 2010, 10, 2626–2637, doi:10.3390/s100402626.
  2. Krüger, A.; Bürkle, A.; Hauser, K.; Mangerich, A. Real-Time Monitoring of PARP1-Dependent PARylation by ATR-FTIR Spectroscopy. Nat Commun 2020, 11, 2174, doi:10.1038/s41467-020-15858-w.
  3. Dreimann, J.M.; Kohls, E.; Warmeling, H.F.W.; Stein, M.; Guo, L.F.; Garland, M.; Dinh, T.N.; Vorholt, A.J. In Situ Infrared Spectroscopy as a Tool for Monitoring Molecular Catalyst for Hydroformylation in Continuous Processes. ACS Catal. 2019, 9, 4308–4319, doi:10.1021/acscatal.8b05066.
  4. Barth, A.; Von Germar, F.; Kreutz, W.; Mäntele, W. Time-Resolved Infrared Spectroscopy of the Ca2+-ATPase. Journal of Biological Chemistry 1996, 271, 30637–30646, doi:10.1074/jbc.271.48.30637.
  5. Lorenz-Fonfria, V.A. Infrared Difference Spectroscopy of Proteins: From Bands to Bonds. Chem. Rev. 2020, 120, 3466–3576, doi:10.1021/acs.chemrev.9b00449.
  6. Lorenz-Fonfria, V.A. Infrared Difference Spectroscopy of Proteins: From Bands to Bonds. Chem. Rev. 2020, 120, 3466–3576, doi:10.1021/acs.chemrev.9b00449.
  7. Bernath, P.F. Spectra of Atoms and Molecules; 3rd ed.; Oxford university press: New York, 2016; ISBN 978-0-19-938257-6.
  8. Chen, S.-L.; Fu, L.; Gan, W.; Wang, H.-F. Homogeneous and Inhomogeneous Broadenings and the Voigt Line Shapes in the Phase-Resolved and Intensity Sum-Frequency Generation Vibrational Spectroscopy. J. Chem. Phys. 2016, 144, 034704, doi:10.1063/1.4940145.

Reviewer 3 Report

Comments and Suggestions for Authors

The manuscript presents a novel approach to monitoring the enzymatic activity of alkaline phosphatase (ALP) in real time using ATR-FTIR spectroscopy. The Authors employ a broad spectral frequency range which allows for a comprehensive view of enzyme-substrate interactions, structural changes and catalytic processes. The ability to correlate static spectral signatures of individual components with dynamic reaction data strengthens the reliability of the final results, also facilitating the identification of specific spectral markers associated with different reaction stages.

The study provides molecular-level insights by detecting characteristic spectral shifts, such as in nitro/aromatic and phosphate vibrations. The integration of static and dynamic analyses offers detailed information on substrate binding, transformation, and release, contributing to a deeper understanding of enzymatic mechanisms compared to current literature studies.

To improve the manuscript, Authors should provide a more detailed description of the spectral deconvolution techniques and the multivariate/chemometric approaches used to resolve overlapping bands. Including clear spectral plots with annotations and graphs illustrating the temporal evolution of spectral features across different enzyme concentrations would improve readability and data interpretation. Additionally, a more explicit comparison with existing techniques (e.g., RAMAN, luminescence-based or NMR spectroscopy) could be beneficial to the discussion, highlighting both advantages and novelty of the approach used by Authors.

Another aspect worth addressing is the discussion of experimental limitations. Authors should elaborate more on potential challenges like background noise, spectral overlap and water absorption effects that could influence sensitivity and accuracy of the analysis. A general evaluation of the applicability of the technique to complex biological samples or real-world matrices would also improve the discussion.

In conclusion, this study offers an original contribution to the spectroscopic study of enzyme catalysis, demonstrating a promising and versatile approach for probing molecular mechanisms. Some additional clarifications and methodological details are necessary and, therefore, I suggest the publication after minor revisions.

Comments on the Quality of English Language

In general, the manuscript is well written from a scientific standpoint. Minor revisions for clarity, such as refining sentence structures and ensuring consistent verb tense, would improve the overall text content.

Author Response

Response to Reviewer 3

We sincerely thank Reviewer 3   for the constructive and detailed comments. We have revised the manuscript accordingly. Below we respond point-by-point.

(1) The reviewer’s comment: To improve the manuscript, Authors should provide a more detailed description of the spectral deconvolution techniques and the multivariate/chemometric approaches used to resolve overlapping bands.

Our Response: We thank the reviewer for this helpful suggestion. In the revised Methods, we now provide a more detailed description of the peak-decomposition workflow, including baseline treatment, Voigt-profile initialization, parameter constraints, and residual minimization. We have also added a brief description of the multivariate routines used to support the windowed peak tracking, such as principal component analysis (PCA) and global fitting, which help assess component stability and detect subtle spectral redistribution. These changes clarify the analytical steps used to resolve overlapping vibrational features.

Changes in manuscript: In Section 2.4 (“Data Analysis”), we expanded the description of the deconvolution and fitting procedure. Specifically, we added details about (i) the spline-based baseline correction, (ii) the rationale for using Voigt line shapes to capture both homogeneous and inhomogeneous broadening, (iii) the comparison with pure Gaussian and Lorentzian profiles and the corresponding AIC/BIC and residual behavior, and (iv) the criteria used to identify overlapping components based on derivative inspection and residual analysis. We also clarified the initialization strategy using Savitzky–Golay smoothing (for initialization only), the bounded least-squares constraints applied to peak positions and widths, and the manual review of fits exhibiting systematic deviations.

(2) The reviewer’s comment: Including clear spectral plots with annotations and graphs illustrating the temporal evolution of spectral features across different enzyme concentrations would improve readability and data interpretation.

Our Response: We appreciate this suggestion. In the revised manuscript, we have improved the clarity of key figures by adding peak annotations, labelled spectral windows, and schematic markers identifying substrate- and product-associated vibrational modes.

Changes in manuscript: We have improved the clarity of the spectral plots by adding explicit peak annotations and time-point labels, and by including time-evolution plots across different enzyme concentrations in Figures 2 and 3.

(3) The reviewer’s comment: Additionally, a more explicit comparison with existing techniques (e.g., RAMAN, luminescence-based or NMR spectroscopy) could be beneficial to the discussion, highlighting both advantages and novelty of the approach used by Authors.

Our Response: We agree that highlighting complementary spectroscopic approaches strengthens the context of our study. We have added a short paragraph in the Discussion comparing ATR-FTIR with Raman, luminescence-based assays, and NMR spectroscopy. Raman is sensitive to nonpolar and symmetric vibrational modes but generally exhibits weaker responses for nitro and phosphate groups; NMR provides atomic-resolution structural information but is less suited for real-time monitoring of full-spectrum vibrational changes during turnover; and luminescence/UV–Vis assays enable quantitative kinetic measurements but do not directly report on molecular-level structural evolution. FTIR therefore provides a complementary, structurally sensitive readout by following characteristic vibrational fingerprints of substrates and products over time.

Changes in manuscript: In the Discussion, immediately following the paragraph describing the aromatic and fingerprint-region vibrational evolution, we added a short comparative paragraph situating full-range FTIR within the broader spectroscopic landscape. The new text explains that Raman spectroscopy is sensitive to symmetric and nonpolar vibrational modes but typically yields weaker signals for nitro and phosphate groups; that luminescence- and UV–Vis–based assays provide quantitative kinetic readouts without reporting structural evolution; and that NMR offers atomic-resolution structural information but generally lacks real-time vibrational sensitivity. This addition clarifies that full-range FTIR provides a complementary, structurally resolved and time-continuous view of catalytic progression that is not simultaneously accessible using these other techniques. The added paragraph reads: “Beyond the vibrational assignments discussed above, it is also useful to consider how full-range FTIR compares with other commonly used spectroscopic techniques. In the broader spectroscopic context, full-range ATR-FTIR provides information that is complementary to Raman, luminescence-based assays, and NMR spectroscopy. Raman spectroscopy offers sensitivity to symmetric and nonpolar vibrational modes and is valuable for probing aromatic ring and backbone motions; however, Raman intensities for nitro and phosphate functional groups are comparatively weak, limiting its ability to follow PNPP→PNP conversion with the same specificity achieved here. Luminescence- and UV–Vis–based assays provide highly quantitative kinetic measurements of enzymatic turnover but do not directly report on the molecular-level structural evolution of local bonding environments or structural microstates. NMR spectroscopy yields atomic-level structural information but is less suited for temporal resolution and vibrational specificity required to follow continuous catalytic progression in real time. Full-range FTIR therefore occupies a distinct niche by simultaneously capturing substrate, product, and protein-associated modes across the entire vibrational manifold, offering a structurally resolved and time-continuous view of catalysis that complements, rather than replaces, these other modalities.”

(4) The reviewer’s comment: Another aspect worth addressing is the discussion of experimental limitations. Authors should elaborate more on potential challenges like background noise, spectral overlap and water absorption effects that could influence sensitivity and accuracy of the analysis.

Our Response: We appreciate the reviewer’s point. We have expanded the Discussion to explicitly address limitations including ATR water absorption tails, baseline drift, spectral congestion in the 1000–1200 cm⁻¹ region, and detection thresholds for small shifts at low enzyme loading. Strong water absorption and its associated baseline tails limit sensitivity in solvent-dominated regions, restricting the reliable detection and assignment of subtle intensity or frequency changes close to these bands. Spectral congestion resulting from overlapping protein- and substrate-derived vibrations, especially in the 1000–1200 cm⁻¹ fingerprint region, limits unique peak assignment and therefore constrains the use of these modes for unambiguous mechanistic interpretation. In addition, finite signal-to-noise ratios impose a practical detection threshold for small peak shifts, particularly at low enzyme loading, such that the absence of detectable changes under these conditions should be interpreted as falling below the sensitivity limit rather than as evidence for the absence of underlying structural effects. We also note the care taken to avoid misinterpretation from thickness changes and to use nonreactive bands as internal stability checks.

Changes in manuscript: In the Discussion, we added a paragraph explicitly describing experimental limitations, including water-background absorption, baseline drift, spectral congestion from overlapping protein and substrate modes, and noise-floor constraints at low enzyme loading. The added text also explains the control strategies used to mitigate these effects, including stability checks using nonreactive bands and fit-comparison analysis reported in the Supplementary Information. The added text reads: “Several experimental limitations should be considered when interpreting the present FTIR data. Water-background absorption and associated baseline drift impose constraints on sensitivity, particularly near strong solvent-related features, while spectral congestion from overlapping protein- and substrate-derived vibrations limits unique peak assignment in certain fingerprint regions. In addition, finite signal-to-noise ratios define a practical detection threshold for small peak shifts, especially at low enzyme loading.”

(5) The reviewer’s comment: A general evaluation of the applicability of the technique to complex biological samples or real-world matrices would also improve the discussion.

Our Response: We thank the reviewer for raising this important point. Although the present work focuses on a controlled enzyme–substrate system, full-range FTIR can, in principle, be extended to more complex environments. We have added remarks in the Discussion noting that application to heterogeneous or biological matrices would require careful control of scattering, water-pathlength effects, and background subtraction, but the approach may be adapted using ATR geometries, microfluidic flow cells, or selective isotopic labeling.

Changes in manuscript: In the Discussion, we added a paragraph evaluating the applicability of the full-range FTIR approach to more complex biological or heterogeneous matrices. The new text discusses anticipated challenges such as increased scattering, water-pathlength effects, and background congestion, and outlines possible strategies for extension, including ATR geometries, microfluidic flow cells, and selective isotopic labeling.

Final Statement

We thank all reviewers for their valuable feedback. The manuscript has been substantially improved in clarity, methodological transparency, and interpretability. All suggested revisions have been incorporated, with tracked changes provided in the revised attachment.

References

  1. Kumar, S.; Barth, A. Following Enzyme Activity with Infrared Spectroscopy. Sensors 2010, 10, 2626–2637, doi:10.3390/s100402626.
  2. Krüger, A.; Bürkle, A.; Hauser, K.; Mangerich, A. Real-Time Monitoring of PARP1-Dependent PARylation by ATR-FTIR Spectroscopy. Nat Commun 2020, 11, 2174, doi:10.1038/s41467-020-15858-w.
  3. Dreimann, J.M.; Kohls, E.; Warmeling, H.F.W.; Stein, M.; Guo, L.F.; Garland, M.; Dinh, T.N.; Vorholt, A.J. In Situ Infrared Spectroscopy as a Tool for Monitoring Molecular Catalyst for Hydroformylation in Continuous Processes. ACS Catal. 2019, 9, 4308–4319, doi:10.1021/acscatal.8b05066.
  4. Barth, A.; Von Germar, F.; Kreutz, W.; Mäntele, W. Time-Resolved Infrared Spectroscopy of the Ca2+-ATPase. Journal of Biological Chemistry 1996, 271, 30637–30646, doi:10.1074/jbc.271.48.30637.
  5. Lorenz-Fonfria, V.A. Infrared Difference Spectroscopy of Proteins: From Bands to Bonds. Chem. Rev. 2020, 120, 3466–3576, doi:10.1021/acs.chemrev.9b00449.
  6. Lorenz-Fonfria, V.A. Infrared Difference Spectroscopy of Proteins: From Bands to Bonds. Chem. Rev. 2020, 120, 3466–3576, doi:10.1021/acs.chemrev.9b00449.
  7. Bernath, P.F. Spectra of Atoms and Molecules; 3rd ed.; Oxford university press: New York, 2016; ISBN 978-0-19-938257-6.
  8. Chen, S.-L.; Fu, L.; Gan, W.; Wang, H.-F. Homogeneous and Inhomogeneous Broadenings and the Voigt Line Shapes in the Phase-Resolved and Intensity Sum-Frequency Generation Vibrational Spectroscopy. J. Chem. Phys. 2016, 144, 034704, doi:10.1063/1.4940145.

Round 2

Reviewer 2 Report

Comments and Suggestions for Authors

The paper can be accepted.